# Towards Next-Generation Sequencing (NGS)-Based Newborn Screening: A Technical Study to Prepare for the Challenges Ahead

**DOI:** 10.3390/ijns8010017

**Published:** 2022-02-24

**Authors:** Abigail Veldman, Mensiena B. G. Kiewiet, Margaretha Rebecca Heiner-Fokkema, Marcel R. Nelen, Richard J. Sinke, Birgit Sikkema-Raddatz, Els Voorhoeve, Dineke Westra, Martijn E. T. Dollé, Peter C. J. I. Schielen, Francjan J. van Spronsen

**Affiliations:** 1Division of Metabolic Diseases, Beatrix Children’s Hospital, University Medical Center Groningen, University of Groningen, 9713 GZ Groningen, The Netherlands; f.j.van.spronsen@umcg.nl; 2Department of Genetics, University Medical Center Groningen, University of Groningen, 9713 GZ Groningen, The Netherlands; r.j.sinke@umcg.nl (R.J.S.); b.sikkema01@umcg.nl (B.S.-R.); 3Department of Laboratory Medicine, University Medical Center Groningen, University of Groningen, 9713 GZ Groningen, The Netherlands; m.r.heiner@umcg.nl; 4Department of Human Genetics, Radboud University Medical Center, 6500 HB Nijmegen, The Netherlands; marcel.nelen@radboudumc.nl (M.R.N.); Dineke.Westra@radboudumc.nl (D.W.); 5Centre for Health Protection, National Institute for Public Health and the Environment, 3720 BA Bilthoven, The Netherlands; els.voorhoeve@rivm.nl (E.V.); martijn.dolle@rivm.nl (M.E.T.D.); 6Centre for Population Screening, National Institute for Public Health and the Environment, 3720 BA Bilthoven, The Netherlands; peter.schielen@isns-neoscreening.org

**Keywords:** next-generation sequencing, first-tier, heel prick, dried blood spot, inherited metabolic disorders, inborn errors of metabolism, newborn screening

## Abstract

Newborn screening (NBS) aims to identify neonates with severe conditions for whom immediate treatment is required. Currently, a biochemistry-first approach is used to identify these disorders, which are predominantly inherited meta1bolic disorders (IMD). Next-generation sequencing (NGS) is expected to have some advantages over the current approach, for example the ability to detect IMDs that meet all screening criteria but lack an identifiable biochemical footprint. We have now designed a technical study to explore the use of NGS techniques as a first-tier approach in NBS. Here, we describe the aim and set-up of the NGS-first for the NBS (NGSf4NBS) project, which will proceed in three steps. In Step 1, we will identify IMDs eligible for NGS-first testing, based on treatability. In Step 2, we will investigate the feasibility, limitations and comparability of different technical NGS approaches and analysis workflows for NBS, eventually aiming to develop a rapid NGS-based workflow. Finally, in Step 3, we will prepare for the incorporation of this workflow into the existing Dutch NBS program and propose a protocol for referral of a child after a positive NGS test result. The results of this study will be the basis for an additional analytical route within NBS that will be further studied for its applicability within the NBS program, e.g., regarding the ethical, legal, financial and social implications.

## 1. Introduction

The aim of newborn screening (NBS) is to identify neonates in whom immediate treatment is life-saving, significantly improves health outcomes, or reduces morbidity. Worldwide, NBS programs have proven their value for many disorders. These are predominantly inherited metabolic disorders (IMDs) with a monogenetic origin, but genetic non-IMDs, e.g., congenital adrenal hyperplasia, cystic fibrosis (CF) and severe combined immunodeficiency (SCID), are screened for in the Netherlands and many other countries as well.

With the exception of SCID screening, the current Dutch NBS program is mainly performed using a biochemistry-first approach. For the biochemical screening, dried blood spots (DBS) are collected between 72 and 168 h after birth and analyzed for abnormal biochemical marker concentrations or enzyme activities [1]. Despite the success of NBS, the approach has some limitations. Although its false positive rates are low compared to other population-based screenings [2,3,4,5], several disorders in NBS, such as classic galactosemia [6], tyrosinemia type 1 [7] and isovaleric acidemia [8], still have a relatively high false positive rate. This results in unnecessary referrals and parental concern. In addition, some IMDs, like urea cycle defects such as N-acetylglutamate synthase deficiency (OMIM 237310) and carbamoyl phosphate synthetase 1 deficiency (OMIM 237300), cannot be included in NBS because the existing biomarkers are most likely not sufficiently sensitive and specific [9].

The use of next-generation sequencing (NGS) as a first-tier test might overcome both issues. Today, NGS is used as a first-tier test in diagnostics within neonatal intensive care units for children who are suspected of having an IMD or other genetic disease [10,11,12,13] and is an effective method to detect genetic defects in severely ill neonates. Furthermore, studies in which whole exome sequencing (WES) was performed to confirm diagnosed IMDs in DBS from patients have shown promising opportunities for NGS in NBS [14,15,16], including a reduction of false positive results in the case of CF screening [17].

The use of NGS has additional advantages. It is easier to include a disorder in screening by adding a gene to a genetic panel than by adding a validated biochemical test to a biochemical panel. Moreover, adding genes to a virtual panel (i.e., one created by digitally filtering whole exome data for a set of regions of interest) for the analysis of WES/whole-genome sequencing (WGS) data is easier than adding them to a targeted panel. There is also a need to include more IMDs in NBS because these—often very rare—disorders remain difficult to diagnose, even as more of them become treatable [18,19]. Furthermore, for some disorders like isovaleric aciduria, it is possible to distinguish between severe and mild phenotypes, since specific genetic variants are associated with severe versus mild symptoms [20]. Notably, as some pathogenic variants related to isovaleric aciduria are located in the introns, special attention should be paid to covering these variants in NGS methods.

However, the actual step of applying NGS analysis as a first-tier approach in screening laboratories has not been taken, although theoretical reviews and pilot studies have explored some of the possibilities and challenges of NGSf4NBS [15,16,21,22,23,24,25]. A major concern is that NGS, and more specifically WES, is not specific or sensitive enough to be a first-tier screen for most IMDs [16]. For instance, in a pilot study with 15 patients diagnosed with IMDs, Boemer et al. [14] showed that it was difficult to interpret genetic variants due to fluctuations in coverage over implicated regions. Only a few studies have tested the use of NGS as a first-tier approach in NBS. Adhikari et al. [16] performed a large retrospective study in the US and found that WES had an overall lower sensitivity and specificity compared to the standard biochemical method. However, the sensitivity varied with disorder, in line with results from another large prospective study [22]. These data show that there are major challenges that have to be resolved before implementation of NGS as a first-tier approach in NBS.

Therefore, the aim of our study is to investigate the use of NGS techniques as a first-tier in the present Dutch NBS system, and not per se to replace the current biochemical NBS for genetic disorders. We have now received a grant (ZONMW 50-54300-98-506) to technically explore the possibilities of NGS in NBS for two years. In three steps, we will work from selecting disorders with an apparent clinical phenotype and developing a preferred technical approach toward the first preparations for the implementation of NGS in the national NBS program. As IMDs comprise the larger part of the screening panel, we will focus on this group of disorders. This article reports on the design of our study in order to highlight the challenges ahead, both at a national and international level, while acknowledging the specific circumstances of the Dutch NBS system.

## 2. Materials and Methods

### 2.1. General Outline of the Study

In this project, we will explore the technical challenges to NGSf4NBS in three steps (Figure 1). Step 1 aims to identify IMDs eligible for NGS-first testing in NBS using a systematic literature review and discussions among experts. In Step 2, we will develop a rapid NGS-based workflow for NBS (including an analytical and bioinformatic pipeline) for the selected IMD-related genes. This will establish feasibility, and we will compare different technical approaches in terms of speed, robustness, sensitivity, specificity and costs in order to establish best-practices and quality standards. Finally, in Step 3, we will prepare for testing the NGS-first strategy integrated into the workflow of a screening laboratory and for an implementation study of the incorporation of this pipeline in the existing Dutch NBS program. We will also propose a protocol for the referral of a child after a positive NGS test result and hope to take a first look into the ethical, practical and financial consequences of NGS-based NBS. Each step is explained in more detail below.

### 2.2. Methods to Identify Which IMDs Are Eligible for NGS-First Testing (Step 1)

The criterium “treatability” from the Wilson and Junger criteria [26,27,28] was previously used in the decision-making about which diseases to include in the current Dutch NBS program [29,30]. Therefore, in NGSf4NBS, we decided that treatability will also be the main selection criterium to identify IMDs eligible for NGS-first testing. We consider an IMD treatable when early intervention substantially improves health outcome. For the selection process, we will start with the list of all currently described IMDs according to the International Classification of Inherited Metabolic Disorders (ICIMD) [31]. This ICIMD report published by Ferreira et al. [31] describes 1459 different IMDs and their associated causative genes. The first selection will be based on a study of van Karnebeek et al. [32] in which approximately 480 IMDs that present with intellectual developmental delay in the clinic were selected to review their treatability. In total, 89 IMDs were classified as treatable and the remaining 391 IMDs as non-treatable [32]. After re-assessment to confirm the absence of new treatment possibilities, non-treatable diseases will be excluded from the ICIMD list. The remaining list will be used as a starting point to systematically assess the treatability of each IMD, following pre-established inclusion and exclusion criteria (Figure 2). This systematic literature review will be conducted by three independent reviewers, and the final selection for diseases eligible for NGS will be evaluated by a panel of experts. To ensure that the final selection is broadly accepted in the field, this panel includes pediatricians and other colleagues with known experience in IMDs from the Dutch Advisory Committee Neonatal Screening for IMDs and the NGSf4NBS study group.

### 2.3. Developing a Rapid NGS-Based Workflow for NBS (Step 2)

Next, we will determine the most suitable NGS platform by focusing on technical aspects, including isolation of DNA from DBS cards, the best analytical approach and a suitable analysis pipeline. Three NGS methods will be considered: a targeted gene panel approach and WES or WGS with a virtual panel.

#### 2.3.1. Samples

A total of 62 DBS cards will be made available by the metabolic departments of Dutch University Medical Centers (UMCs). These DBS will be derived from IMD patients diagnosed via NBS or later in life after symptom development. Based on the list of treatable IMDs from Step 1, we will assemble a representative selection of samples. An effort will be made to include samples with a known genetic variant, as much as possible, so they can be confirmed in this experiment. We will include different types of variants, like single nucleotide variants, copy number variants and insertion/deletion variants (indels), so that we can test the performance of the different techniques for the range of variants that are expected to be detected with NGS. Some variants in genes with pseudogenes will be included as well. Samples will only be included when parental consent for use for research purposes is registered in the research registries of the UMCs. The Ethical Review Board (ERB) of the University Medical Centre Groningen has declared that this study does not fall under the Medical Research Involving Humans Act, so no further approval of the ERB is needed (202100550).

#### 2.3.2. DNA Isolation

DNA isolation from DBS cards is feasible [33,34]. We will use a previously developed automated method that was tested in our laboratory. To use as little material as possible, DNA will be extracted from one blood spot (approximately 50 µL) [17], after which DNA will be isolated on a solid support from the Protocol IQ Casework Pro Kit for Maxwell 16 (Promega, Madison, WI, USA) (IQ) [35]. If needed, we will also test other DNA extraction methods.

#### 2.3.3. NGS Methods

Next, we will investigate which NGS method is optimal in an NBS setting. We will compare a target-enrichment system with WES and WGS. We will develop a state-of-the-art custom-designed targeted panel for the IMDs selected in Step 1 that allows targeted and thorough sequencing of relevant regions of interest only. Different target-enrichment methods will be compared theoretically, in consultation with experienced partners, and the final choice will depend on the location of the target regions of the causative (likely) pathogenic (LP/P) variants in the selected IMDs and the expected performance when using DNA from DBS [36].

In a first pilot testing phase, 12 DBS samples of patients diagnosed with an IMD with a known causative genetic variant will be sequenced with the targeted panel and with WES followed by the use of a virtual panel to see whether any analytical hurdles are encountered. After assessing the general quality of the data generated by both methods, this first experiment will provide an indication of the feasibility of detecting the genetic variants present in the tested samples. It will also provide information on the coverage of all genes of interest and indicate possible limitations of the different methods. Subsequently, all three NGS approaches—WES, WGS, and the targeted panel—will be tested on 50 DBS cards of IMD patients. Next to checking the coverage of all genes of interest and the detection of already known variants in the samples, we will also compare the methods on other criteria relevant for a screening setting, e.g., throughput time of the laboratory analytical work-up, flexibility of the process and financial costs. Depending on the results of these experiments, we will determine the best technical approach.

#### 2.3.4. Data Analysis

We will compare the number of LP/P variants and variants of unknown significance (VUS) in both the targeted NGS and WES/WGS approaches, without knowledge of the phenotype and clinical features of the tested individuals. Therefore, it is important to determine how the NGS analysis and variant interpretation pipeline should be organized in order to identify only relevant variants, as much as possible. The pipelines that will be used for data analysis at the University Medical Centre Groningen and Radboud University Medical Center have already been validated for diagnostic purposes [37,38]. Reference transcripts will be derived from the National Center for Biotechnology Information RefSeq database (http://www.ncbi.nlm.nih.gov/RefSeq/, accessed on 8 January 2022). The default reference will be selected based on clinical relevance, but the position of the detected variant and its effect on protein sequence on all relevant transcripts will be investigated during analysis. To optimize the balance between finding disease-causing pathogenic variants and non-pathogenic variants, we will test both custom-designed, in-house-developed (University Medical Centre Groningen and Radboud University Medical Center) and commercially available (i.e., Franklin (franklin.genoox.com)) bioinformatics approaches. In general, data filtering by in-house-developed methods will follow the same steps as other diagnostic NGS panels [37,39]. We will aim first for a method that allows us to detect all known LP/P variants, i.e., 100% sensitivity for LP/P variants. We will then optimize the data filtering strategy to detect as few VUSes and (likely) benign variants as possible without losing the required sensitivity. As all samples are anonymized and no AMCG-59 list genes [40] or other genes that might harbor an unsolicited finding [41] are in the virtual panel, there is no chance of identifying an unsolicited finding.

### 2.4. Preparing for the Incorporation of the Pipeline into the Dutch NBS Program (Step 3)

Collaboration between diagnostic clinical laboratories and partners of the Dutch NBS program brings together the unique expertise of NGS-based genetics and active screening programs. Therefore, the last part of the project aims at developing a decision tree for the selection of the variants that will be reported, including formulating expert opinions on how to (re)evaluate or possibly (biochemically) validate VUSes. We will formulate recommendations for handling of unsolicited findings using the existing Dutch national guidelines of the Dutch Society for Laboratory specialists clinical genetics (VKGL) and the Association of Clinical Genetics Netherlands (VKGN) [41] as a starting point. We will also propose a procedure for the referral and follow-up of the newborns with a pathogenic outcome. The decision tree will be integrated into the pipeline to generate a fully automated workflow. Once the pipeline has successfully been tested by the diagnostic partners in Step 2, it will be tested in the routing of the Dutch NBS program using the same samples and settings. By repeating the sequencing of the same samples in a different setting, we can assess the reproducibility and applicability of the initial results while also building NGS expertise within the Dutch NBS program. We plan to conclude this project by giving an overview of the practical and financial consequences that will be encountered when implementing an NGS (first) strategy in Dutch NBS. Notably, we fully acknowledge the need for automation of this process, and the study aims to develop a strategy that enables a quick decision about whether a newborn should be referred to a pediatric metabolic specialist or whether the result should have an appropriate IMD-specific follow-up. Since the Dutch NBS system deals with hundreds of DBS samples a day, the entire process from DBS sampling to NGS result and referral should be fast, error-free and efficient.

## 3. Discussion

Current NBS screening is vital for children’s health, and continuous improvements are indispensable for providing state-of-the-art healthcare. Thus, the development and implementation of appropriate new techniques is crucial. Here, we present the design of an explorative and technical study into the application of NGS in NBS, possibly in an NGS-first fashion. While we recognize that the use of NGS as a first-tier in NBS is still a long way from (inter)national implementation, we believe it is time to explore the possibilities and limitations of this technique in NBS, as laid out in the current study proposal.

In Step 1 of this project, IMDs eligible for NGS-first testing will be selected. In Figure 2, we present the study flow that will be used. We selected treatability as the first criterium based on Wilson and Junger’s criteria that “There should be an accepted treatment for patients with recognized disease.” [26,27,28]. However, a disease for which a treatment has been reported sporadically [21,23,32] will not be considered to fulfill this criterium. It is to be expected that, besides the IMDs currently included in the Dutch NBS, some IMDs and causative genetic variants included in other large studies exploring NGS-techniques [16,21,22,23,24] will be selected in Step 1, for example argininosuccinate lyase deficiency (ASL) and Wilson’s disease (WD). We acknowledge that eventually all Wilson and Junger criteria must be covered to include disorders in NGS-based neonatal screening. Step 2 concerns the development of a rapid NGS-based workflow for NBS. Here, we will compare three NGS methods (WES/WGS and a targeted panel) and establish an analysis pipeline. The performance and feasibility of the chosen workflow will be tested for all three methods using a test set of 50 IMD DBS samples. Designing and establishing this screening method covers Wilson and Junger’s criterium 5 by contributing to a reliable screening method [26,27,28]. Besides technical aspects of the different approaches, many other practical issues need to be considered. An important aspect of selecting the most suitable technique for NBS is its flexibility. With an eye on the future, we would like to have the possibility to expand the number of disorders covered by the screening, preferably without creating an entirely new panel or pipeline, as treatment options are developing at an ever increasing pace. The main pitfall of a targeted approach consisting of a fixed set of genes is that the complete panel would need to be updated and re-evaluated when new genes need to be added. A WES/WGS approach with a bioinformatic filter for the genes identified in Step 1 is thus much more flexible and can easily be expanded.

Additionally, since the clinical use of NGS has exponentially increased [10,11,12,13,42], costs have gradually declined [43]. Both WES and WGS may therefore become attractive for NBS both financially and because of their adaptability [14], although WGS will continue to be relatively expensive for the next few years. Whether WGS is worth the extra costs, for example to detect new intronic pathogenic variants not covered in WES, needs to be addressed. The exact throughput time of NGS analysis in a screening setting is also still an open question. However, obtaining a result within four days seems feasible (Figure 3), which would mean it can be used within the timeframe of current NBS if sampling is performed within 124 h after birth. To make NBS even faster in the future, the use of umbilical cord blood collected directly after birth has been suggested [44,45], necessitating additional sampling in the Dutch NBS program.

NGS, especially WES/WGS, will provide large amounts of data that will need to be analyzed and interpreted correctly. Most IMDs are caused by various genetic variants, and the detected variants are sometimes discordant with the clinical diagnosis [46,47]. Moreover, with the use of NGS techniques, new disease-causing variants or VUS might be encountered. Translating the data into information meaningful for referral requires a strategy for the interpretation of variants that enables fast follow-up. In addition, the genetic data need to be available to return to if more information becomes known about that child. Compared to the known diagnostic situation for patients who present with clinical symptoms, two circumstances are uniquely important for NGS-first. First, at the time of screening, we do not have information on the clinical phenotype of the child. Secondly, the throughput time should be as short as possible because treatment should be started as soon as possible. Different automated and semiautomated tools have been developed to interpret genetic variants causing different kinds of disorders, including IMDs [47,48], to aid these decisions. In our experience, the different types of genetic databases used for interpretation of genes and variants also provide different information. Therefore, application of several genetic databases is crucial to deal with genetic variants of IMDs in NGS.

Despite the advantages, the use of NGS as a first-tier alternative is still questioned in literature [15,16,25], especially as the current biochemistry-based NBS is highly successful, with a sensitivity of 98% and a specificity of 99.865% in the Netherlands [49]. We have to realize that, especially for IMDs with low and very low incidences (e.g., for 3-methylcrotonyl-CoA carboxylase deficiency, very long-chain acyl-CoA dehydrogenase deficiency and carnitine transporter deficiency/carnitine uptake defect) [16], it will be some time before enough data is available to test whether NGS techniques can reach the same level of performance as biochemical screening. However, for CF [15,50,51] and SMA, there is already evidence that NGS as first-tier testing could be promising [15]. At least in a pilot setting, NGS could be used in parallel to the biochemical methods to test the same sample and thus provide support for the initial outcome and reduce false-positive referrals. The effectiveness of this second-tier approach has been shown by Adhikari et al. [16,25]. Together, these data show that thorough investigation is needed to find an approach in which ultimately the most benefits can be gained from applying NGS in NBS, regardless of whether this is as a first-tier, second-tier, or parallel approach.

In Step 3 of our project, we will build a method that can be used in an implementation study for the Dutch NBS system. Here, there is a need to make choices and to address issues like security of data, legal consequences and overall costs. Few genomic data need to be stored with a targeted approach. In contrast, with a WES/WGS approach, the entire genomic profile may be stored from the beginning of a person’s life. Examples of extensive genomic databases like deCODE genetics Iceland show that holding this data has numerous ethical implications, e.g., how to deal with the stored information on carriership of the breast cancer genes *BRCA-1* and *BRCA-2* and whether to inform and screen affected individuals [52,53,54,55]. In an implementation trajectory, privacy and genomic data protection have to be ensured by strict governmental legislation. Public views thus have to be taken into account when genomic data are stored in population databases and used.

In this context, another Wilson and Junger criterium may be most important: ‘The test should be acceptable for the population.’ [26,27,28] Current NBS is highly accepted in society. At present, the level of participation in NBS in Europe is higher than 90% and, in many countries, higher than 99% [56]. This high level of acceptance is in danger if a screening test is implemented for which there are still uncertainties about the meaning of the test result and its value for the screened population [57,58]. Thus, NGSf4NBS should be implemented only when there is no doubt about its effectiveness, accuracy, security and added value compared with the current NBS programs, and communities should be informed about the advantages, risks and challenges of the NGSf4NBS. A lack of confidence in this screening test in the general population will lead to a reduced acceptance of NBS, which is something that must be prevented at all costs [59,60].

## 4. Conclusions

In short, NGSf4NBS is a technical study that will work toward a proof-of-principle application of NGS in NBS in a relatively short period of time. It will deliver valuable information on the feasibility of an NGS in NBS approach. As the technique has now been embraced by diagnostics, it is unlikely that this momentum will be lost and thus, inevitably, NGS will find some place in NBS. However, it might take some more time before it will be fully adopted. In the words of Sydney Brenner: ‘Progress in science depends on new techniques, new discoveries and new ideas, probably in that order.’ [61]. We acknowledge that previous expansions have shown that implementation into a complex program like NBS takes time and careful consideration together with NBS stakeholders. It currently takes years before all the possible issues following the addition of a single new disorder in NBS are resolved. We therefore anticipate it will take as much or substantially more time to resolve all issues when introducing a group of diseases detected by NGS. Whatever the method chosen in NBS, the entire process, from heel prick to referral and diagnostics results in the hospital, should be efficient and swift, necessitating optimal collaboration of the complete Dutch NBS chain with both the preventive and curative healthcare system. Through our project, we are attempting to take NGS in NBS a step further to prepare for the future in collaboration with these important stakeholders.

## Figures and Tables

**Figure 1 IJNS-08-00017-f001:**
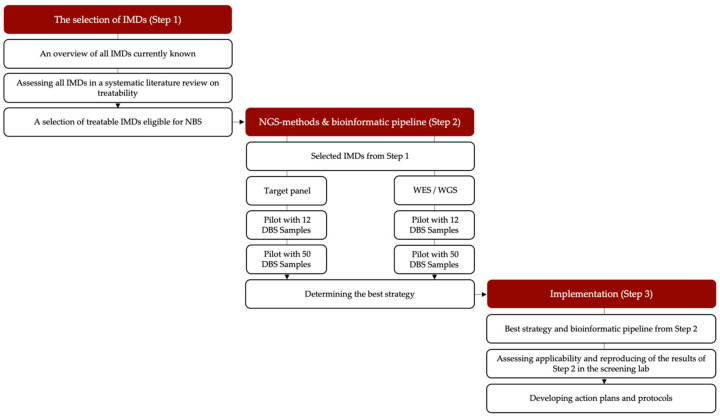
Overview of the study set-up. IMD = inherited metabolic disorder, NGS = next-generation sequencing, DBS = dried blood spot, WES/WGS: whole exome sequencing/whole genome sequencing.

**Figure 2 IJNS-08-00017-f002:**
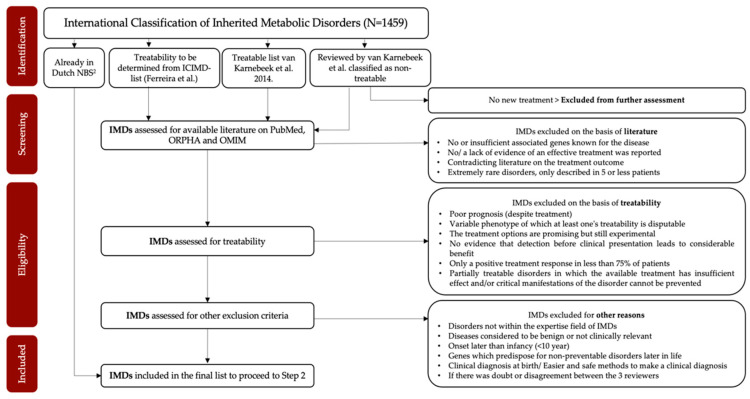
Preliminary flowchart of the selection process of Step 1.

**Figure 3 IJNS-08-00017-f003:**
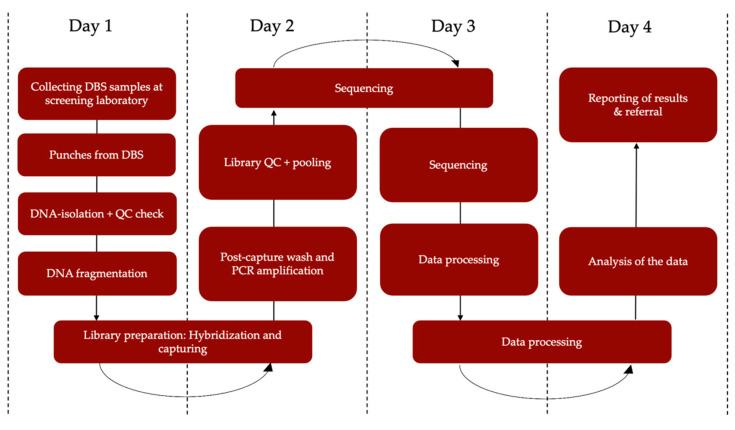
Estimated turn-around-time from NGS to result. DBS = dried blood spot, QC = quality control, PCR = polymerase chain reaction.

## Data Availability

Not applicable.

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
