# Peer review of "Towards Next-Generation Sequencing (NGS)-Based Newborn Screening: A Technical Study to Prepare for the Challenges Ahead"

_2409-515X, 2022, doi:10.3390/ijns8010017_

Round 1

Reviewer 1 Report

Summary

The authors describe the overall structure of their NGSf4NBS study which will explore the use of next-generation sequencing (NGS) as a first-tier newborn screening (NBS) test. The manuscript is well structured and clear. As the manuscript is an outline of a study that is not yet completed, the information in the manuscript is general and lacks specific details some of which will determined during the study. However, some specifics should be added to the manuscript especially in the methodology, so that the reader knows exactly how and what data the NGSf4NBS study will be using when making decisions.  

Specific comments/questions –

Line 71 – you mention genes can be easily added. It is important to clarify here, that it is easy to add genes to a virtual panel using WES/WGS data, but less so for a targeted enrichment panel; you correctly highlight this fact in the discussion starting on line 264, but clarification in the introduction could benefit the reader.

Line 143 –  “NBSf4NGS” used here, while “NGSf4NBS” used elsewhere.

Line 162 – “we will mainly include…”, will include other variant types too? If so, which?

Line 163 – “inversion/deletion variants (indels)”, this should be insertion/deletion variants.

Line 174 – please check the reference for the extraction developed in your lab. The listed reference does not seem to contain a DNA extraction method.

Line 193 – “assessing” should be “assess”

Line 206 – “…both approaches…”, are you referring to targeted NGS and WES/WGS here?

General comments/questions –

  • It is unclear if the 12 DBS pilot samples are part of the 50 total samples obtained.
  • “Bio-informatics” should be changed to “bioinformatics” throughout.
  • Will the NGS pipeline utilized be analytically validated? Please define any pipeline validation in the methods, e.g. GIAB samples [https://pubmed.ncbi.nlm.nih.gov/30858580/].
  • You touch on the ethical issues surrounding incidental NGS findings in the discussion (BRCA), but is there a formal plan for the management of incidental findings? https://pubmed.ncbi.nlm.nih.gov/23788249/
  • Will reportable variants be subject to orthogonal validation? If so, does the turn-around time mentioned on line 275 account for this?
  • More details need to be added for how you will go from selected IMDs to a list of genes.
    • Will you use existing curated gene panels? Something like PanelApp [https://www.nature.com/articles/s41588-019-0528-2]. Expert collaboration and consensus on clinical gene panels has the potential to benefit patient outcomes [https://www.sciencedirect.com/science/article/abs/pii/S0002929721002688] or will selected panel of experts create a gene list from the selected disorders?
    • Selection of transcript is important, particularly so for targeted gene vs WES/WGS. See https://www.ncbi.nlm.nih.gov/pmc/articles/PMC7335342/ - If the NGSf4NBS study group will select genes, details of which transcripts will be selected should be provided, i.e. LRG (https://www.lrg-sequence.org/) , MANE (https://www.ncbi.nlm.nih.gov/refseq/MANE/) etc.
  • For variant classification, I see mention of Franklin, which uses ACMG/AMP criteria in scoring variants, but it should be explicitly stated what criteria/processes you will use to classify variants and what scoring system will be used to determine the optimal balance mentioned on line 212. For example, what is one system detects 2 known disease-causing variants, but also identifies 20 VUS, while another only detects 1 known disease-causing variant and no VUS; are you optimizing for sensitivity, specificity or some other measure?
  • A formal plan for reevaluation of variants should be established or discussed if already established. Reevaluation has the potential to impact diagnoses, see https://www.sciencedirect.com/science/article/pii/S1098360021053600 and https://www.ahajournals.org/doi/10.1161/CIRCGEN.120.003047

Author Response

Dear editor,

We are very grateful for being given the opportunity to resubmit our manuscript “Towards Next Generation Sequencing (NGS)-based newborn screening. Heads up: technical study to prepare for the challenges ahead”. We would like to thank you and the reviewers for their careful evaluation. Their comments have been very useful. The manuscript has now been revised according to their suggestions and a point-by-point reply to reviewers’ questions is provided. We hope that you will deem the revised manuscript suitable for publication in the International Journal of Neonatal Screening.

Reviewer 1

  1. The authors describe the overall structure of their NGSf4NBS study which will explore the use of next-generation sequencing (NGS) as a first-tier newborn screening (NBS) test. The manuscript is well structured and clear. As the manuscript is an outline of a study that is not yet completed, the information in the manuscript is general and lacks specific details some of which will determined during the study. However, some specifics should be added to the manuscript especially in the methodology, so that the reader knows exactly how and what data the NGSf4NBS study will be using when making decisions.  

Response: We thank the reviewer for the compliments. We do agree that the manuscript would benefit from adding more details about the methodology, and we have added this information in the revised version of this manuscript, especially in the paragraphs describing data analysis and preparation for implementation on page 6 of the manuscript. For more details we refer the reviewer to the answers on more specific questions below.

Specific comments/questions

  1. Line 71 – you mention genes can be easily added. It is important to clarify here, that it is easy to add genes to a virtual panel using WES/WGS data, but less so for a targeted enrichment panel; you correctly highlight this fact in the discussion starting on line 264, but clarification in the introduction could benefit the reader.

Response: We thank the reviewer for this suggestion and agree that the introduction benefits from adding this extra information. We have now added a sentence on page 2 lines 70-74.

  1. Line 143 –  “NBSf4NGS” used here, while “NGSf4NBS” used elsewhere.

Response: We apologize for this typo and have corrected it. We have also checked the rest of the manuscript for the same mistake.

  1. Line 162 – “we will mainly include…”, will include other variant types too? If so, which?

Response: In our study we aim to add different variant types, including SNV’s, CNV’s and indels in order to test the performance of different NGS techniques on a range of variants known to exist in the genes we aim to screen. We will focus on these types of variants and have now clarified the sentence on page 5 lines 176-177.

  1. Line 163 – “inversion/deletion variants (indels)”, this should be insertion/deletion variants.

Response: We have corrected the typo.

  1. Line 174 – please check the reference for the extraction developed in your lab. The listed reference does not seem to contain a DNA extraction method.

Response: We apologize for this mistake, we indeed refer to the wrong article regarding the DNA isolation method  we will use, and deleted it. As mentioned in the paragraph DNA will be isolated on a solid support from the Protocol IQ Case- 176 work Pro Kit for Maxwell 16 (Promega, Madison Wisconsin, USA) (IQ), following the manufacturer’s instructions.

7. Line 193 – “assessing” should be “assess”)

Response: We have slightly changed the sentence in order to make it grammatically correct.

8. Line 206 – “…both approaches…”, are you referring to targeted NGS and WES/WGS here?

Response: Yes, we mean targeted NGS and WES/WGS here. We have now added this information to the text on page 5 line 220 to clarify this in the manuscript,

General comments/questions –

  1. It is unclear if the 12 DBS pilot samples are part of the 50 total samples obtained.

Response: We understand that it was not clear if the 12 DBS pilot samples were part of the 50 samples. We have now replaced ’50 samples’ in total on page 5 line 171 by ‘62’, in order to clarify that we used 12 samples in the first step, and 50 samples in the second step, so a total of 62 different samples.

  1. “Bio-informatics” should be changed to “bioinformatics” throughout.

Response: We have now made this change throughout the manuscript.

  1. Will the NGS pipeline utilized be analytically validated? Please define any pipeline validation in the methods, e.g. GIAB samples [https://pubmed.ncbi.nlm.nih.gov/30858580/].

Response: The pipelines used in this study are part of the diagnostic workflow at the University Medical Centre Groningen and Radboud University Medical Centre in Nijmegen. Validation of these pipelines is a standard procedure in the diagnostics setting, which is performed every time a new version of the pipeline is put into use. The  pipelines in Groningen and Nijmegen have been published by Corsten‐Janssen et al. [1] and Haer-Wigman et al. [2] respectively. We have now included this information  (including the references) on page 6 lines 227-229 of the manuscript.

  1. You touch on the ethical issues surrounding incidental NGS findings in the discussion (BRCA), but is there a formal plan for the management of incidental findings? https://pubmed.ncbi.nlm.nih.gov/23788249/

Response: Although the chance of an incidental finding in this study is small, because we use a gene panel of 100 genes related to metabolic  disorders,  we indeed cannot rule out completely an incidental finding.

In this study only fully anonymized samples are analyzed, preventing potential incidental findings to be reported to the patients. However, we do agree with the reviewer that incidental findings are an important issue that needs consideration before NGS can be implemented in NBS. Therefore, in Step 3 of the project we will address this topic. We will formulate recommendations for handling of incidental findings, for which we will use the existing Dutch national guidelines of the Dutch Society for Laboratory specialists clinical genetics (VKGL) and the Association of Clinical Genetics Netherlands (VKGN) as a starting point for discussions [3]. We have now included a sentence on incidental findings during the study on page 6 lines 242-244 of the manuscript, and emphasized that it will be part of the discussions in Step 3 of the project on page 6 lines 251-255.

  1. Will reportable variants be subject to orthogonal validation? If so, does the turn-around time mentioned on line 275 account for this?

Response: In our study we will analyze DBS samples with previously detected LP/P variants. The classification of the previously detected variants is according to diagnostic rules, based a.o. on co-segregation, presence of the variant in several affected individuals or functional studies. In our cohort of 50 samples, we expect to also detect VUSes. For these VUSes no additional test will be performed to allow reclassification of these variants, because the aim of our study is screening for known (likely) pathogenic variants. However, we agree with the reviewer that in order to prepare for implementation of NGS in NBS, it is important to develop a strategy on handling VUSes. If we will decide that VUSes will be included in further analyses in the screening program, then a clear protocol regarding follow up of VUSes is needed since we agree with the reviewer that this will increase turn-around time. Therefore, this topic will be further addressed by the experts working on Step 3 of the project, and we will formulate recommendations on the best strategy. We have included a sentence in which we emphasize that the topic of how to handle will be addressed in Step 3 of the project on page 6 lines 251-255 in the manuscript.

  1. More details need to be added for how you will go from selected IMDs to a list of genes.

Will you use existing curated gene panels? Something like PanelApp [https://www.nature.com/articles/s41588-019-0528-2]. Expert collaboration and consensus on clinical gene panels has the potential to benefit patient outcomes [https://www.sciencedirect.com/science/article/abs/pii/S0002929721002688] or will selected panel of experts create a gene list from the selected disorders?

Response: The list of genes has been selected by experts in the field of metabolic disorders. We used the International Classification of Metabolic Disorders as published in 2021 by Ferreira et al. [4] As a starting point for the selection of disorders, we received an Excel file from prof. dr. Ferreira with 1450 IMDs . Disorders included in this list were classified based on their genetic background, which means that the causative gene was leading in the definition of a certain disorder, instead of their clinical presentation, and in this way only one gene is related to a specific disorder. This way of classification made the selection process of genes straightforward. However, it should be noted that this has consequences for the nomenclature. For example, different degrees of disease severity are listed as separate entities and not subtypes. In cases of locus heterogeneity (multiple genes associated with the same phenotype) the involvement of each gene was considered as the basis for inclusion..

We have now included this information in the manuscript at page 4 lines 142-143.

  1. Selection of transcript is important, particularly so for targeted gene vs WES/WGS. See https://www.ncbi.nlm.nih.gov/pmc/articles/PMC7335342/ - If the NGSf4NBS study group will select genes, details of which transcripts will be selected should be provided, i.e. LRG (https://www.lrg-sequence.org/) , MANE (https://www.ncbi.nlm.nih.gov/refseq/MANE/) etc.

Response: We agree with the reviewer that selection of the proper transcript is crucial for interpretation of the detected variants. In this study, the reference will be derived from the National Center for Biotechnology Information RefSeq database (http://www.ncbi.nlm.nih.gov/RefSeq/). According to the ACMG guidelines, the transcript representing either the longest known transcript and/or the most clinically relevant transcript based on tissue specific expression will be selected as the default reference sequence. During analysis of data from both the targeted panel and WES, the position and effect on protein sequence of the detected variant will be checked for all transcripts, and the variant will be interpreted for each interesting transcript. We have now added this information in the manuscript on page 6 lines 229-233.

  1. For variant classification, I see mention of Franklin, which uses ACMG/AMP criteria in scoring variants, but it should be explicitly stated what criteria/processes you will use to classify variants and what scoring system will be used to determine the optimal balance mentioned on line 212. For example, what is one system detects 2 known disease-causing variants, but also identifies 20 VUS, while another only detects 1 known disease-causing variant and no VUS; are you optimizing for sensitivity, specificity or some other measure?

Response: In general, data filtering will follow the same steps as described for other diagnostic NGS panels [5]. Briefly, quality filtering of the called variants was performed. Artefacts and polymorphisms were excluded based on our in-house list of ‘managed variants’, as well as variants with a high allele frequency in cohorts of healthy controls (ExAC, gnoMAD and GoNL), and known LB/B variants. For classification, we will use the Dutch shared database [6] where known (LP/P) variants are already labeled as such. Variants without a label will be classified following diagnostic guidelines [7], based on ACMG criteria. In our study we include 50 DBS samples with known LP/P variants. We will first aim for a method which allows us to detect all known (likely) pathogenic variants, so a sensitivity of 100% (for these LP/P variants). In case of a relatively high number of VUS/LB/B variants left, we can adjust thresholds for a higher specificity. Based on our experience in this project, in Step 3 of the project, where we make the first steps to implementation of NGS in NBS, we will formulate a recommendation for the optimal data filtering strategy. We now mention this in the manuscript on page 6 lines 237-242.

  1. A formal plan for reevaluation of variants should be established or discussed if already established. Reevaluation has the potential to impact diagnoses, see https://www.sciencedirect.com/science/article/pii/S1098360021053600 and https://www.ahajournals.org/doi/10.1161/CIRCGEN.120.003047

Response: Re-evaluation of variants is not primarily the goal of this study or screening program. Here, we aim to develop a NGS- based procedure for NBS. We will analyze DBS samples with known LP/P variants previously detected by Sanger sequencing, which makes the chance of finding a variant which will be re-classified as (likely) pathogenic  small. However, we will  collect VUSes left after data-filtering. In Step 3 of the project, where we will start preparing for implementation of NGS in NBS, we will address this topic in a broader perspective and formulate a recommendation on how to handle the VUSes in NBS. Regarding this topic we also refer the reviewer to the answers given to question 13 and 16.

 REFERENCES

  1. Corsten‐Janssen N, Bouman K, Diphoorn JCD, Scheper AJ, Kinds R, Mecky J, et al.. A prospective study on rapid exome sequencing as a diagnostic test for multiple congenital anomalies on fetal ultrasound. Prenatal Diagnosis 2020;40(10):1300–9
  2. Haer-Wigman L, van Zelst-Stams WA, Pfundt R, van den Born LI, Klaver CC, Verheij JB, et al. Diagnostic exome sequencing in 266 Dutch patients with visual impairment. Eur J Hum Genet. 2017;25:591–9.
  3. Van Der Schoot V, Haer-Wigman L, Feenstra I, Tammer F, Oerlemans AJM, Van Koolwijk MPA, et al.. Lessons learned from unsolicited findings in clinical exome sequencing of 16,482 individuals. European Journal of Human Genetics 2021
  4. Ferreira CR, Rahman S, Keller M, Zschocke J, Abdenur J, Ali H, et al.. An international classification of inherited metabolic disorders ( ICIMD ). Journal of Inherited Metabolic Disease 2021;44(1):164–77
  5. Alimohamed MZ, Johansson LF, Posafalvi A, Boven LG, Van Dijk KK, Walters L, et al.. Diagnostic yield of targeted next generation sequencing in 2002 Dutch cardiomyopathy patients. International Journal of Cardiology 2021;332:99–104
  6. Fokkema IFAC, Velde KJ, Slofstra MK, Ruivenkamp CAL, Vogel MJ, Pfundt R, et al.. Dutch genome diagnostic laboratories accelerated and improved variant interpretation and increased accuracy by sharing data. Human Mutation 2019;40(12):2230–8
  7. Weiss MM, Van Der Zwaag B, Jongbloed JDH, Vogel MJ, Brüggenwirth HT, Lekanne Deprez RH, et al.. Best Practice Guidelines for the Use of Next-Generation Sequencing Applications in Genome Diagnostics: A National Collaborative Study of Dutch Genome Diagnostic Laboratories. Human Mutation 2013;34(10):1313–21Yours sincerely,
    Abigail Veldman

Reviewer 2 Report

This is a manuscript describing a protocol for introducing NGS as a first-tier test in NBS. It is highly relevant to discuss such introduction as thoroughly discussed and argumented for in the manuscript. The NBS society will need to consider such introduction and find ways to do this. The protocol described by Veldman et al. is sound, well-described and relevant, and the methods proposed seems also doable in the time-frame and national context in which they are to be done. The methods are relevantly discussed in a clear language. While the topic is highly relevant to the NBS society, my main concern is whether the IJNS has a history and willingness to accept such a purely protocol-manus – it certainly is not noted in the types of manus accepted – could perhaps fit into technical note, in which case a re-working of the manus will be necessary.

I do not have comment or critism to the protocol a such – it is sound. I probably would have spend a little more time on discussing the relevance of introducing parallel biochemical screening and would also include it in the screening algoritm, as there probably will be a number of VUS that will need biochemical clarification; one should not think that doing biochemistry in parallel would increase time spend. The sentence starting line 223, I am a little unsure about – please make more clear.

Author Response

Dear editor,

We are very grateful for being given the opportunity to resubmit our manuscript “Towards Next Generation Sequencing (NGS)-based newborn screening. Heads up: technical study to prepare for the challenges ahead”. We would like to thank you and the reviewers for their careful evaluation. Their comments have been very useful. The manuscript has now been revised according to their suggestions and a point-by-point reply to reviewers’ questions is provided. We hope that you will deem the revised manuscript suitable for publication in the International Journal of Neonatal Screening.

Reviewer 2

  1. This is a manuscript describing a protocol for introducing NGS as a first-tier test in NBS. It is highly relevant to discuss such introduction as thoroughly discussed and argumented for in the manuscript. The NBS society will need to consider such introduction and find ways to do this. The protocol described by Veldman et al. is sound, well-described and relevant, and the methods proposed seems also doable in the time-frame and national context in which they are to be done. The methods are relevantly discussed in a clear language. While the topic is highly relevant to the NBS society, my main concern is whether the IJNS has a history and willingness to accept such a purely protocol-manus – it certainly is not noted in the types of manus accepted – could perhaps fit into technical note, in which case a re-working of the manus will be necessary.

Response: We thank this reviewer for his valuable comments. As to the form of the paper, we have checked with the managing editor of IJNS whether IJNS accepts protocol based manuscripts and got confirmation that IJNS does. We have chosen to submit our manuscript as a ‘Communication’ as we intended to write a concise paper that needs to be published quickly. The manuscript type ‘Project Report’ may also fit-if the Editor of IJNS prefers to submit our manuscript as a Project Report, we are happy to do so.

  1. I do not have comment or critism to the protocol a such – it is sound. I probably would have spend a little more time on discussing the relevance of introducing parallel biochemical screening and would also include it in the screening algoritm, as there probably will be a number of VUS that will need biochemical clarification; one should not think that doing biochemistry in parallel would increase time spend.

Response: Thank you for your comment. Biochemical confirmation is particularly of interest for VUSes, increasing the sensitivity of a genetics-first approach. As this study particularly investigates the technical feasibility of genetic screening, parallel biochemical screening is beyond the scope of this project. However, we agree with the reviewer that in order to prepare for implementation of NGS in NBS, it is important to develop a strategy on follow up of VUSes. If we decide that VUSes will be included in further analyses in the screening program, then a clear protocol regarding follow up of VUSes is needed. Therefore, this topic will be further addressed by the experts working on Step 3 of the project, and we will formulate recommendations on the best strategy. We have included a sentence in which we emphasize that the topic of how to handle will be addressed in Step 3 of the project on page 6 lines 251-255 in the manuscript.

  1. The sentence starting line 223, I am a little unsure about – please make more clear.

Response: We agree with the reviewer that this sentence is confusing, and we have clarified it.

Yours sincerely,
Abigail Veldman

Reviewer 3 Report

The Communications manuscript submitted by Veldman et al describe a technical study design that will investigate the feasibility of using next-generation sequencing technologies as a first tier screening approach for inherited metabolic disorders. While I understand the exploratory nature of the proposed study, it would be interesting to at least include more detail about proposed experimental design, examples of inherited metabolic disorders the authors intend to study, examples of variant panels, and details on the bioinformatic tools that will be used for analysis would be most useful information to include in the body of the test. As the manuscript stands now, it is written as a vague proposal than a proposed study design that would be interesting for the audience to learn about.

Additionally, extensive editing is required for fix sentence and syntax structure that is proper for the English language. 

Specific comments:

Figure 1: IMD is defined differently in the figure (inborn metabolic disorder) as compared to how it is defined within the body of the text (inherited metabolic disorder).

Line 140 - It would be interesting to see some preliminary results of the systematic review, how the process works, list a few of the diseases that would be eligible for the proposed study design, and discuss briefly if there have been any published NGS methods for those diseases.

Line 157 - I question if 50 DBS cards will be a sufficient sample size to obtain a significant idea of whether the NGS process would work. Especially, as written in Line 134, there are 89 total IMDs that were classified as treatable.

Line 190 - Again, it would be good to note a few example IMDs that the authors intend to include in their study and the known causative genetic variant that they intend to interrogate via sequencing. 

Line 192 - The term "virtual panel" is vague and not defined properly here.

Line 275 - If the authors are anticipating a turn-around time of 4 days, it may be useful to include a figure that outlines how they intend to reach this turn-around time and how long they envision each step to take. 

Author Response

Dear editor,

We are very grateful for being given the opportunity to resubmit our manuscript “Towards Next Generation Sequencing (NGS)-based newborn screening. Heads up: technical study to prepare for the challenges ahead”. We would like to thank you and the reviewers for their careful evaluation. Their comments have been very useful. The manuscript has now been revised according to their suggestions and a point-by-point reply to reviewers’ questions is provided. We hope that you will deem the revised manuscript suitable for publication in the International Journal of Neonatal Screening.

Reviewer 3

  1. The Communications manuscript submitted by Veldman et al describe a technical study design that will investigate the feasibility of using next-generation sequencing technologies as a first tier screening approach for inherited metabolic disorders. While I understand the exploratory nature of the proposed study, it would be interesting to at least include more detail about proposed experimental design, examples of inherited metabolic disorders the authors intend to study, examples of variant panels, and details on the bioinformatic tools that will be used for analysis would be most useful information to include in the body of the test. As the manuscript stands now, it is written as a vague proposal than a proposed study design that would be interesting for the audience to learn about.

Response: We thank this reviewer for his valuable comments. MDPI-journals, IJNS included, accept Project reports, communications and protocols as article types. We have chosen to submit our manuscript as a ‘Communication’ as we intended to write a concise paper that needs to be published quickly. The manuscript type ‘Project Report’ may also fit-if the Editor of IJNS prefers to submit our manuscript as a Project Report, we are happy to do so. We hope that with the careful revision of the combined comments of all reviewers, we have eliminated any remaining vagueness. Please find below a careful and detailed response to your additional point-by-point critique.

  1. Additionally, extensive editing is required for fix sentence and syntax structure that is proper for the English language. 

Response: After finishing the revised manuscript we have send it to an native English speaking editor to check and correct the English language. We trust that this aspect of the manuscript has now improved.

Specific comments:

  1. Figure 1: IMD is defined differently in the figure (inborn metabolic disorder) as compared to how it is defined within the body of the text (inherited metabolic disorder).

Response: We apologize for this inconsistency, and we have now updated figure 1.

  1. Line 140 - It would be interesting to see some preliminary results of the systematic review, how the process works, list a few of the diseases that would be eligible for the proposed study design, and discuss briefly if there have been any published NGS methods for those diseases.

Response: We have to the best of our ability described the process of selection of treatable diseases eligible for inclusion in Step 1 of the project on page 4 of the manuscript and in Figure 2. Unfortunately, at the moment we do not have any results available on this part of the project. However, we expect that besides the metabolic disorders already included in the Dutch neonatal screening that will be selected directly (Figure 2 of the manuscript), it is likely that disorders that were used in other large studies investigating NGS-techniques, such as NEXUS [8], the Babyseq project [9-11] and Adhikari et a. [12] will be in our results as well. We added a sentence about this expectation together with a few of these disorders as an example in the discussion on page 7 lines 290-294 of the manuscript

  1. Line 157 - I question if 50 DBS cards will be a sufficient sample size to obtain a significant idea of whether the NGS process would work. Especially, as written in Line 134, there are 89 total IMDs that were classified as treatable.

Response: The aim of Step 2 of the project is to develop the technical procedures to a NGS-first NBS to the extent that it is technically and analytically ready for a development trajectory in the NBS reference laboratory of the Dutch Institute National Health and Environment (RIVM). It is a first step in developing an approach, in which different techniques (targeted panel, WES and WGS) will be tested and compared. We chose, in order to make an honest comparison, to use the same 50 samples (62 including the pilot samples) for all three techniques instead of 150 different samples. We believe that analyzing this amount of DBS will give us enough data to select the best method for working with DBS samples regarding feasibility, limitations, speed, and costs of the methods, which are based on workflows which are already used in a diagnostics setting. We will make a selection of DBS containing a range of interesting variant types like single nucleotide variants, copy number variants and insertion/deletion variants (indels) to investigate which can be detected with NGS methods. The reviewer correctly noticed that we will probably classify more than 50 disorders as treatable. However, we believe that assessing the performance of NGS-based NBS screening, and possibly comparing it to the current biochemical methods, for each included disorder is out of the scope of this project, but will have to be part of a later implementation trajectory of the RIVM.

  1. Line 190 - Again, it would be good to note a few example IMDs that the authors intend to include in their study and the known causative genetic variant that they intend to interrogate via sequencing. 

Response: For this question we would also like to refer the reviewer to the answer to question 4. Unfortunately, at the moment we do not have any results available on this part of the project. However, we expect that besides the metabolic disorders already included in the Dutch neonatal screening that will be selected directly (Figure 2 of the manuscript), it is likely that disorders that were used in other large studies investigating NGS-techniques, such as NEXUS [1], the Babyseq project [2-4] and Adhikari et al. [5] will be in our results as well. We added a sentence about this expectation together with a few of these disorders as an example in the discussion on page 7 lines 290-294 of the manuscript

  1. Line 192 - The term "virtual panel" is vague and not defined properly here.

Response: We have now added the definition of a virtual panel, digitally filtering of a gene panel a sequencing the whole exome, on page 2 lines 72-73, where the word ‘virtual panel’ is used for the first time in the manuscript.

  1. Line 275 - If the authors are anticipating a turn-around time of 4 days, it may be useful to include a figure that outlines how they intend to reach this turn-around time and how long they envision each step to take. 

Response: We thank the reviewer for the nice suggestion to add a figure in order to clearly indicate the foreseen workflow of NGS in NBS and the corresponding time frame. We have now included this figure in the manuscript (figure 3).

REFERENCES

  1. Milko LV, O'Daniel JM, Decristo DM, Crowley SB, Foreman AKM, Wallace KE, et al. An Age-Based Framework for Evaluating Genome-Scale Sequencing Results in Newborn Screening. The Journal of Pediatrics. 2019;209:68-76
  2. Holm IA, Agrawal PB, Ceyhan-Birsoy O, Christensen KD, Fayer S, Frankel LA, et al. The BabySeq project: implementing genomic sequencing in newborns. BMC Pediatrics. 2018;18(1)
  3. Ceyhan-Birsoy O, Murry JB, Machini K, Lebo MS, Yu TW, Fayer S, et al. Interpretation of Genomic Sequencing Results in Healthy and Ill Newborns: Results from the BabySeq Project. The American Journal of Human Genetics. 2019;104(1):76-93
  4. Wojcik MH, Zhang T, Ceyhan-Birsoy O, Genetti CA, Lebo MS, Yu TW, et al. Discordant results between conventional newborn screening and genomic sequencing in the BabySeq Project. Genetics in Medicine. 2021
  5. Adhikari AN, Gallagher RC, Wang Y, Currier RJ, Amatuni G, Bassaganyas L, et al. The role of exome sequencing in newborn screening for inborn errors of metabolism. Nature Medicine. 2020;26(9):1392-7

Round 2

Reviewer 3 Report

The authors have appropriately responded to all of my comments from my initial review and I am satisfied with the response.